# Direct Furfural Production from Deciduous Wood Pentosans Using Different Phosphorus-Containing Catalysts in the Context of Biorefining

**DOI:** 10.3390/molecules27217353

**Published:** 2022-10-29

**Authors:** Prans Brazdausks, Daniela Godina, Maris Puke

**Affiliations:** Latvian State Institute of Wood Chemistry, Dzerbenes 27, LV-1006 Riga, Latvia

**Keywords:** lignocellulose sources, biorefining, furfural, phosphorus-containing catalysts, hydrolysis, enzymatic hydrolysis

## Abstract

This study seeks to improve the effectiveness of the pretreatment stage when direct furfural production is integrated into the concept of a lignocellulosic biomass biorefinery. First of all, the catalytic effects of different phosphorus-containing salts (AlPO₄, Ca₃(PO₄)₂, FePO₄, H₃PO₄, NaH₂PO₄) were analysed in hydrolysis for their ability to convert birch wood C-5 carbohydrates into furfural. The hydrolysis process was performed with three different amounts of catalyst (2, 3 and 4 wt.%) at a constant temperature (175 °C) and treatment time (90 min). It was found that the highest amount of furfural (63–72%, calculated based on the theoretically possible yield (% t.p.y.)) was obtained when H₃PO₄ was used as a catalyst. The best furfural yield among the used phosphorus-containing salts was obtained with NaH₂PO₄: 40 ± 2%. The greatest impact on cellulose degradation during the hydrolysis process was observed using H₃PO₄ at 12–20% of the initial amount, while the lowest degradation was observed using NaH₂PO₄ as a catalyst. The yield of furfural was 60.5–62.7% t.p.y. when H₃PO₄ and NaH₂PO₄ were combined (1:2, 1:1, or 2:1 at a catalyst amount of 3 wt.%); however, the amount of cellulose that was degraded did not exceed 5.2–0.3% of the starting amount. Enzymatic hydrolysis showed that such pretreated biomass could be directly used as a substrate to produce glucose. The highest conversion ratio of cellulose into glucose (83.1%) was obtained at an enzyme load of 1000 and treatment time of 48 h.

## 1. Introduction

Lignocellulosic biorefining technologies are expected to become more important for several decades to come. Today, their economic viability is affected by the efficient conversion of all three major constituents of lignocellulosic (LC) biomass (cellulose, hemicellulose, and lignin) into value-added biobased chemicals and biofuels. Despite the fact that the lignocellulosic biorefinery concept has a clear value proposition, commercial success at the industrial scale is still inadequate.

One of the reasons is that lignocellulosic biomass cannot be converted selectively into target products without a loss of valuable components, mainly hemicellulose and cellulose [1,2]. Hemicellulose currently represents the largest polysaccharide fraction wasted from most lignocellulosic biorefineries [3,4]. Therefore, the valorisation of hemicellulose into high-value chemicals may very well be a key solution for overcoming the techno-economical constraints of high operating costs for lignocellulosic biorefineries.

Furfural production is one of the valuable ways to utilise hemicellulose. Furfural itself is an important feedstock for various organic platform chemicals, such as furfuryl alcohol, furoic acid, phenolic resin, etc. [5,6]. Unfortunately, the drawbacks of commercial furfural production methods are becoming a major stumbling block to integrating them into the lignocellulosic biorefinery concept. In addition, the current furfural production process is characterised by low production efficiency. The yield of furfural does not exceed 55% t.p.y. A large proportion (approximately 40–50%) of the starting C-6 carbohydrate content in the solid residue becomes irreversibly degraded [7]. Therefore, in research over the last decade, great efforts have been made to develop novel and efficient furfural production processes [8,9]. Despite the fact that these studies show promising results, there has not yet been achieved a perfect balance between energy efficiency, high furfural yield and economic benefits. Therefore, we are still forced to look for effective ways to produce furfural.

In a previous study [10], we found that it was possible to achieve higher conversion efficiency of pentosans into furfural than today’s furfural production plants (more than 65%, t.p.y.). In addition, more than 90% of the initial cellulose content could be saved in the lignocellulosic residue after furfural production. However, there was a drawback—despite the good properties of phosphoric acid, it needs to be neutralised before the fermentation stage [11,12]. To avoid this treatment stage, the direct use of phosphate salts may be a solution. In addition, phosphate salts can be used as buffers in the fermentation process and as nutrients for microorganisms [13], but there are no reports on the use of phosphate salt hydrolysis to produce furfural as well as its effect on enzymatic hydrolysis.

Therefore, the aim of this study was to determine the effect of phosphorus-containing salts (AlPO₄, Ca₃(PO₄)₂, FePO₄, NaH₂PO₄) as catalysts for the production of furfural from birch wood chips, and to measure the amount of cellulose remaining in the solid lignocellulosic residue for further processes.

## 2. Materials and Methods

### 2.1. Materials and Chemicals

D-(+)-Xylose (≥99%), Cellobiose (≥99%), D-(+)-glucose (≥99%), L-(+)-arabinose (≥99%), D-(+)-mannose (≥99%), D-(+)-galactose (≥99%), formic acid (≥95%), acetic acid (≥99%), levulinic acid (≥98%), 5-hydroxymethylfurfural (≥99%), 2-furaldehyde (≥99%), sulfuric acid (H_2_SO_4_) (95–97%), phosphoric acid (H₃PO₄) (99%), aluminium phosphate (AlPO₄) (≥93%), calcium phosphate (Ca₃(PO₄)₂) (96%), iron(III) phosphate (FePO₄) (96%), and sodium phosphate (NaH₂PO₄) (96%) were purchased from Merck (Darmstadt, Germany) and used without further purification.

### 2.2. Preparation of Feedstock

Birch wood chips from a local plywood production plant, JSC “Latvijas Finieris—Lignums”, were used in this study. These wood chips were a cellulose grade by-product (bark content under 2 wt.%). The moisture content of the collected wood chips was 49 wt.%. To avoid unnecessary wood degradation during storage, the wood chips were air-dried to reduce the moisture content below 15 wt.%. After that, the chips were fractionated using the Muototerä classifier MT300 according to the SCAN-CM 40:01 standard. A sieve arrangement was utilised: oversize chips (45 mm holes) → over thick chips (8 mm slots) → large accept chips (13 mm holes) → small accept chips (7 mm holes) → pin chips (3 mm holes) → fines. A fraction of Ø13–7 mm was used for further experiments.

### 2.3. Characterisation of Feedstock

Before determination of the carbohydrates in the raw feedstock, quantification of the extractives was performed according to the TAPPI 204 cm-07 standard. The used solvent was ethanol–benzene in the ratio 1:2. The extraction time was 6 h. The structural carbohydrates in wood samples before and after furfural production were determined according to the National Renewable Energy Laboratory TP-510-42618 standard (see Table 1). The oven-dried birch wood chips were composed of 37.8% cellulose, 20.7% xylan, 0.5% arabinan, 1.6% galactan, 2.4% mannan, 4.1% acetyl groups, 3.6% acid-soluble lignin, 20.3% acid-insoluble lignin, 4.2% extractives and 0.6% ash.

Concentrations of saccharides were determined via HPLC using a Shimadzu LC20AD liquid chromatograph equipped with a RI detector (Shimadzu RID 10 A) and a Shodex Sugar SP-0810 column at 80 °C, with deionised water as the mobile phase under a flow rate of 0.6 mL/min. The concentrations of hydrolysis process by-products, such as formic acid, acetic acid, levulinic acid, 5-hydroxymethylfurfural and furfural, were also analysed using the same HPLC but with a Shodex Sugar SH-1821 column at 50 °C and 0.008 M H_2_SO_4_ as eluent at a flow rate of 0.6 mL/min. This determination method was also used for analysis of the obtained hydrolysate after phosphoric acid-catalysed hydrolysis.

### 2.4. Hydrolysis Process

The experiments were conducted in a bench-scale reactor system that modelled the industrial furfural obtaining process. The reactor had an internal diameter of 110 mm and a volume of 13.7 dm^3^. The reactor was equipped with a steam jacket and a corresponding automatic control system to maintain constant temperature and pressure throughout the experiment [14] (Figure 1). The hydrolysis time (90 min) and the treatment temperature (175 °C) were constant, while the amount of the catalyst (2, 3 or 4% of oven-dried biomass) was varied. The pressure in the reaction medium was also constant at 7.9 ± 0.1 bar. The used catalysts were AlPO_4_, Ca_3_(PO_4_)_2_, FePO_4_, NaH_2_PO_4_ and H_3_PO_4_, as well as without a catalyst for an autohydrolysis process. To show the catalytic properties of the used chemicals, autocatalytic hydrolysis was performed as a reference process to better understand these salts’ catalytic effects.

The obtained hydrolysates were analysed with the Shimadzu LC20AD HPLC using a Shodex Sugar SH1821 column at 50 °C with 0.008 M sulphuric acid as the mobile phase under a flow rate of 0.6 mL/min. The lignocellulosic residue was dried to ambient moisture content and ground in a Retsch GmbH SM100 cutting mill for chemical analysis, according to the NREL TP-510-42618 standard.

### 2.5. Enzymatic Hydrolysis Process

A preliminary study of enzymatic hydrolysis was also performed to understand the possibilities of using the obtained solid lignocellulosic residue for glucose production. Previously acid-hydrolysed birch wood chips were hydrolysed using the Cellic^®^ CTec2 enzymatic complex (Novozymes). Before enzymatic hydrolysis, these chips were ground in a Retsch GmbH SM100 cutting mill. The used sieve was 0.75 mm. Hydrolysis was performed in duplicate at 50 °C in 0.05 M KH_2_PO_4_–K_2_HPO_4_ buffer at pH 6.7 using a Biosan ES-20/80 orbital shaker at 160 rpm for 12–48 h with a 15% solid load (*w/v*), and the results are presented as the mean. Reducing sugars were determined with the Shimadzu LC20AD HPLC using a Thermo Scientific HyperRez XP Carbohydrate Pb^2^⁺ 8 UM column at 70 °C and deionised water as the mobile phase at a flow rate of 0.6 mL/min.

Enzymatic hydrolysis was performed at different enzyme dosages: 0, 100, 250, 500, 750 and 1000 units/15 g dry substrate. Measurement of enzyme activity was performed according to the method by Denault et al. [15] in combination with the method by Dygert et al. [16]. The calibration curve for glucose determination using the copper (II)–neocuproine method was in the range of 0.01 to 0.1 g/L.

## 3. Results and Discussion

### 3.1. Hydrolysis—Furfural Production

Four different phosphorus-containing salts (AlPO₄, Ca₃(PO₄)₂, FePO₄, NaH₂PO₄) at three different amounts (2, 3 and 4 wt.%, calculated relative to oven-dried mass (o.d.m.)) were used as catalysts during the hydrolysis process to produce furfural from the birch wood chips. Figure 2 demonstrates the visual differences in lignocellulosic residue colour after furfural production with different catalysts. Autocatalysis and H₃PO₄-catalysed hydrolysis were performed as reference processes to better understand these salts’ catalytic effects. The obtained results are summarised in Figure 2.

The yield of furfural produced under autohydrolysis conditions was only 20.7% of the t.p.y. or 3.2 wt.%, calculated relative to o.d.m. Compared with industrial yield in conventional furfural production plants (which does not exceed 55% of the t.p.y.), this yield is approximately 60% lower and the addition of a catalyst is necessary to improve furfural yield. Adding phosphoric acid as a catalyst at 2, 3 and 4 wt.% to the reaction medium, the obtained amount of furfural increased to 62.9, 72.3 and 71.7% of the t.p.y. or 9.7, 11.1 and 11.0% of the o.d.m., respectively (Figure 3). Analysing these experimental data, it can be seen that the highest furfural yield from the birch wood chips was produced at the catalyst amount of 3 wt.%. This allows us to conclude that there is no need to further increase the catalyst amount in the reaction medium. Comparing these results with a previous study [14] where sulfuric acid was used as a catalyst in the same pre-treatment procedure, a similar observation could be made. Unfortunately, the disadvantage is that using phosphoric acid as a catalyst requires a higher temperature to achieve the same reaction efficiency achieved with sulfuric acid at 160 °C [14]. It is also confirmed in other publications [17] that furfural can be produced using phosphoric acid as a catalyst in the temperature range 170–200 °C. On the other hand, when using a greener furfural production process (for example, high-pressure CO_2_ treatment), temperatures higher than 200 °C are needed. Furthermore, an additional step must be included in which hemicellulose is extracted from lignocellulosic biomass before furfural production using the previously mentioned method [18]. From the industrial point of view, this additional step is not desirable.

A different situation can be observed when phosphorus-containing salts are used as catalysts to produce furfural (Figure 3). For example, furfural yield during hydrolysis almost did not change at any amount of AlPO₄ tested—21.3% of the t.p.y. (or 3.3% of the o.d.m.). In addition, furfural yield decreased with an increase in Ca₃(PO₄)₂ amount in the reaction zone. This indicates that Ca₃(PO₄)₂ inhibits birch wood pentosan dehydration and the furfural formation process. This could be due to fact that these salts are not soluble in water. Therefore, they cannot release the cations that initialise the cyclodehydration reaction of pentoses into furfural. As a result, these phosphorus-containing salts are not appropriate as catalysts for the selective separation and conversion of C-5 carbohydrates into furfural.

In contrast, when increasing amounts of FePO₄ or NaH₂PO₄ were added to the reaction zone, both chemicals showed catalytic properties on the conversion of birch wood pentosans into furfural. With an increase in the amount of FePO₄ in the reaction medium, the furfural yield showed a linear increase to 35.0% of the t.p.y., which is two times less than obtained furfural yield under the same hydrolysis conditions when H₃PO₄ was used as a catalyst. Using NaH₂PO₄ as a catalyst obtained the highest furfural yield among the studied phosphorus-containing salts—the furfural yield reached almost 40% of the t.p.y. (or approximately 6% of the o.d.m.). In addition, the obtained results indicated that the effect of the increase in NaH₂PO₄ in the reaction medium displayed similar furfural formation dynamics—at the catalyst amount of 3 wt.%, furfural yield increased, but at the catalyst amount of 4 wt.%, it began to decrease.

Therefore, the general conclusion of the obtained results regarding the effect of AlPO₄, Ca₃(PO₄)₂, FePO₄ and NaH₂PO₄ on the conversion of birch wood pentosans into furfural is that the catalytic properties of these salts are much lower than those of H₃PO₄. Furthermore, an equivalent amount of furfural to today’s industrial furfural production amount (55% of the t.p.y.) could not be obtained by these salts.

### 3.2. Hydrolysis: By-Products Created during the Production of Furfural

When acting on biomass with physicochemical processes, in addition to the target product, by-products are also inevitably formed. In our process formic acid, acetic acid, levulinic acid and 5-HMF were detected as by-products. As can be seen in Figure 4, among these acetic acid was produced with the highest yield (3.5–5.3% of the o.d.m.). Comparing the produced amount of acetic acid during autohydrolysis with the amount produced in the presence of phosphorous-containing salts, it can be concluded that AlPO₄, Ca₃(PO₄)₂ and FePO₄ did not have a catalytic effect on the deacetylation reaction of the acetyl groups. The produced yields of acetic acid were very similar to the yield obtained via autohydrolysis. In contrast, acetic acid was produced at a higher yield than in the autohydrolysis process when using H₃PO₄ and NaH₂PO₄: 5.1–5.3 and 4.2–4.4% of the o.d.m., respectively. Comparing these results with the theoretically possible amount of acetic acid, almost all acetyl groups in the birch wood chips were converted into acetic acid and extracted from the reaction medium. On an industrial scale of furfural production from birch wood chips, the production of acetic acid from the waste streams could therefore be considered. Within the biorefinery concept, such an amount of produced acetic acid indicates that the LC residue after furfural extraction would contain a very small amount of acetyl groups, and thus it would be possible to avoid the formation of an unwanted inhibitor (see Section 3.4).

Other by-products that indicate the destruction of cellulose and furfural during the hydrolysis process, such as formic acid, levulinic acid and 5-HMF, were produced in small quantities (Figure 4).

### 3.3. Hydrolysis: Retained Cellulose

From the point of view of the biorefinery concept, the pretreatment processes’ effects on the amount of cellulose retained in the produced lignocellulosic (LC) residue are also an important indicator. Therefore, the effect of AlPO₄, Ca₃(PO₄)₂, FePO₄, NaH₂PO₄ and H₃PO₄ catalysed hydrolysis, as well as autohydrolysis, on the content of cellulose in LC residue was analysed. The effects of autohydrolysis, H₃PO₄ and phosphorus-containing salts on cellulose degradation during the hydrolysis process are shown in Figure 5.

By comparing the cellulose yield in the LC residue depending on the catalyst amount, it can be seen that the highest amount of cellulose was retained when FePO_4_ was used as a catalyst (97.4% of the initial amount). The second-best result was achieved with NaH_2_PO_4_: 95% of the initial amount. By comparison, although H_3_PO_4_ as a catalyst gave the best results for furfural production, the amount of cellulose retained in the LC residue was the lowest. Therefore, to obtain high furfural yields while retaining a high amount of cellulose in the LC residue, it would be necessary to change the hydrolysis treatment parameters or use different catalysts.

### 3.4. Hydrolysis: Acetyl Groups in the LC Residue

Another important parameter that can affect the effectiveness of biological treatment is the amount of acetyl groups in the obtained LC residue. During biological treatment, acetic acid is formed from these acetyl groups and is considered a critical inhibitor of the enzymatic hydrolysis of cellulose [19]. When analysing the chemical composition of LC residue, the measured amounts of acetic acid produced were recalculated to give the amount of retained acetyl groups. As can be seen in Figure 6, the lowest amount of acetyl groups was found in the LC residue where H_3_PO_4_ was used as a catalyst, with less than 0.4% of the oven-dried LC mass. Twice as many acetyl groups were retained in the LC residue if NaH₂PO₄ was used as a catalyst. These results are consistent with those observed when hydrolysate was analysed after furfural extraction.

Therefore, LC residue that is considerably better suited for enzymatic hydrolysis can be obtained using these two catalysts.

### 3.5. Hydrolysis: Furfural Production Using an H_3_PO_4_/NaH_2_PO_4_ Mixture as a Catalyst

To test our hypothesis, we used a mixture of catalysts (H_3_PO_4_/NaH_2_PO_4_) for furfural production. We tested three different ratios (1:2; 1:1; 2:1). The obtained results are summarised in Table 2. The furfural yield improved and was similar at all ratios of H_3_PO_4_/NaH_2_PO_4_ tested, being higher compared to using NaH_2_PO_4_ alone as a catalyst and lower compared to using H_3_PO_4_ alone. With these catalyst mixtures (the total catalyst amounts were 3 wt.%) the obtained furfural yield was approximately 60% of the t.p.y.

Using the mixture of H_3_PO_4_/NaH_2_PO_4_ as a catalyst, it was possible to preserve more cellulose in the LC residue than if H_3_PO_4_ was used alone. This allows for the conclusion that such a catalytic hydrolysis process is a promising pretreatment stage for integration into a biorefining system where a glucose monomer is used as a feedstock.

### 3.6. Enzymatic Hydrolysis

To determine the suitability of the LC residue obtained in this way for the production of glucose via enzymatic hydrolysis, we chose to work with an LC residue sample treated with an H_3_PO_4_/NaH_2_PO_4_ catalyst mixture at the ratio of 1:1. The cellulose content in this substrate was 45.63% of the o.d.m. The choice to use this LC residue as a substrate was based on the fact that increasing the amount of phosphoric acid in the reaction zone reduces the degree of polymerisation of cellulose [20], while an excessively high concentration of acid inhibits the activity of the enzymes that break down cellulose into glucose monomer over long treatment times [15]. Before performing enzymatic hydrolysis using the Cellic^®^ CTec2 enzyme (Novozymes) and KH_2_PO_4_–K_2_HPO_3_ buffer (pH 6.7), we determined the enzyme activity with the neocuproine method. The determined enzyme activity, according to Denault et al. [15] and Dygert S., et al. [16], was 346 U/mL.

Enzymatic hydrolysis was performed with a fixed load of the enzyme complex (100, 250, 500, 750 and 1000 U) at different reaction times (12, 24, 36 and 48.0 h). As shown in Figure 7, the effect of enzyme load on the effectiveness of the cellulose–glucose conversion ratio was significant in the range 100–500 U at all studied treatment times. When increasing the enzyme load from 500 to 1000 U, the conversion ratio of cellulose into glucose did not increase significantly. This allows for the conclusion that the optimal enzyme load for this type of substrate could be found in the range of 250–750 U. The obtained result also indicates that the treatment time could be reduced to 36 h, because the conversion ratio of cellulose into glucose at 48 h of treatment at all studied enzyme loads was very similar. Based on that, the best conversion ratio of cellulose into glucose (78.8%) can be obtained at 36 h with an enzyme load of 750 U. The highest conversion ratio of cellulose into glucose (83.1%) was obtained at an enzyme load of 1000 and a treatment time of 48 h; the difference in these results is small enough that it is possible to reduce the enzyme load and time, which are crucial parameters from an economic point of view. In addition, the obtained LC residue can be used directly as a substrate for glucose production via enzymatic hydrolysis without rinsing.

During enzymatic hydrolysis, other carbohydrates were also detected in the hydrolysate. Their concentration is summarised in Table 3. It can be seen that regardless of hydrolysis time and enzyme load, the cellobiose, xylose, galactose and arabinose contents were approximately the same and did not significantly increase with increases in enzyme load and treatment time. The biggest difference in content was for glucose, which corresponded to a substrate highly enriched in cellulose.

## 4. Conclusions

This study searched for the best phosphorous-containing catalyst for the co-production of furfural and glucose via catalytic and enzymatic hydrolysis, respectively. It was found that a mixture of NaH₂PO₄ and H₃PO₄ was the best catalyst for the production of furfural, compared to AlPO₄, Ca₃(PO₄)₂, FePO₄, or NaH₂PO₄ or H₃PO₄ alone. Using this catalyst, it was possible to produce approximately 61 ± 2% of the theoretically possible amount of furfural from birch wood and preserve approximately 97 ± 2% of the initially available cellulose in the solid lignocellulosic residue for the production of glucose. It is possible to deduce that this catalyst has selective properties. Preliminary study of glucose production from such pretreated lignocellulosic residue via enzymatic hydrolysis indicated that it was possible to reach a high cellulose–glucose conversion ratio (up to 83% of the theoretically possible amount) in a relatively short time. These findings indicate that it is possible to integrate furfural production into the framework of a lignocellulosic biorefinery, thus opening up the possibility of more efficient use of biomass. Future work must focus on optimisation of the operating conditions of catalytic and enzymatic hydrolysis to maximise the yield of both studied products.

## Figures and Tables

**Figure 1 molecules-27-07353-f001:**
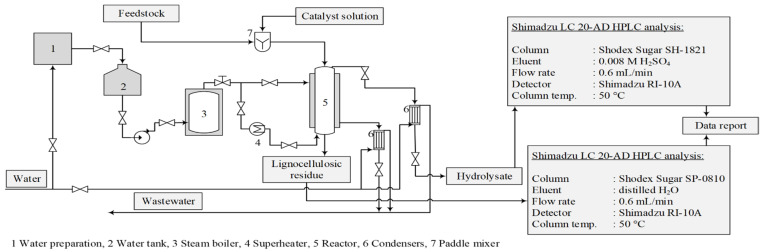
Acid hydrolysis reactor system and experimental procedure for obtaining hydrolysis products.

**Figure 2 molecules-27-07353-f002:**
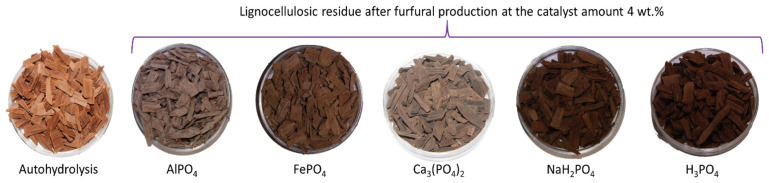
Lignocellulosic residue after furfural production at 4 wt.% of each catalyst.

**Figure 3 molecules-27-07353-f003:**
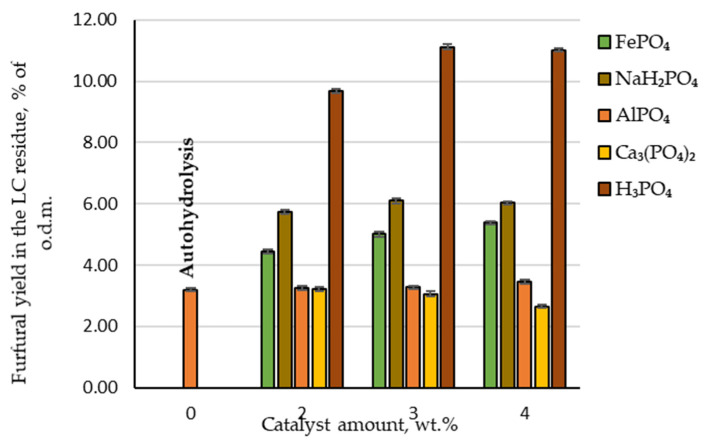
Furfural yields in the lignocellulose residue depending on the catalyst amount.

**Figure 4 molecules-27-07353-f004:**
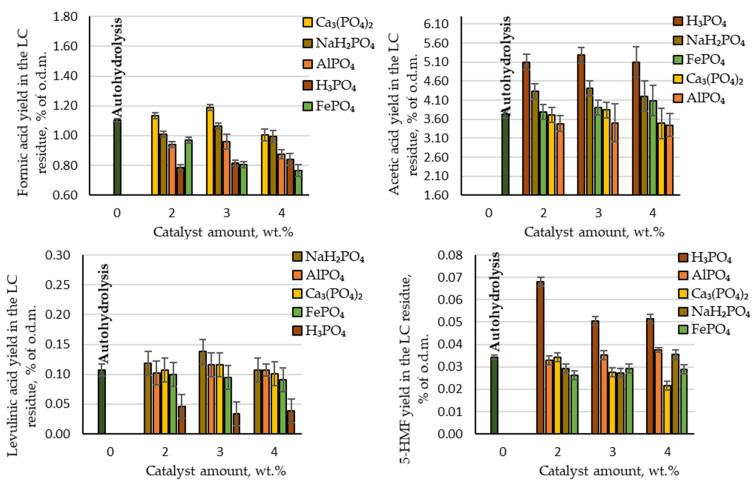
Yields of hydrolysis by-products (formic acid, acetic acid, levulinic acid and 5-HMF) in the LC residue depending on the catalyst amount.

**Figure 5 molecules-27-07353-f005:**
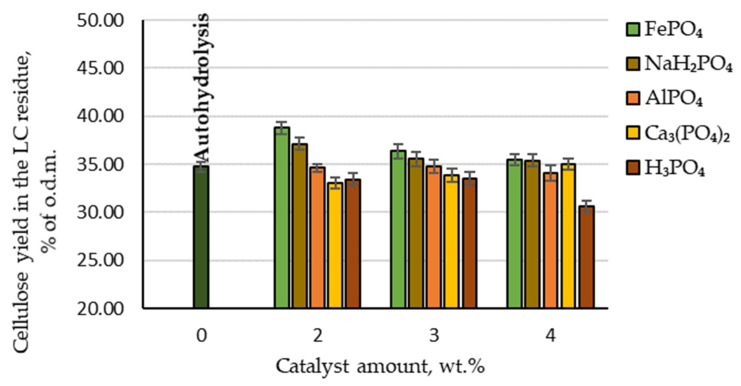
Amounts of cellulose retained in the LC residue depending on the catalyst amount.

**Figure 6 molecules-27-07353-f006:**
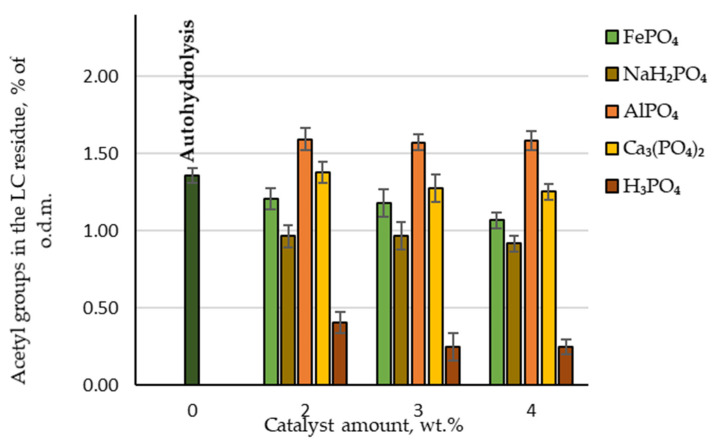
Retained amounts of acetyl groups in the LC residue depending on the catalyst amount.

**Figure 7 molecules-27-07353-f007:**
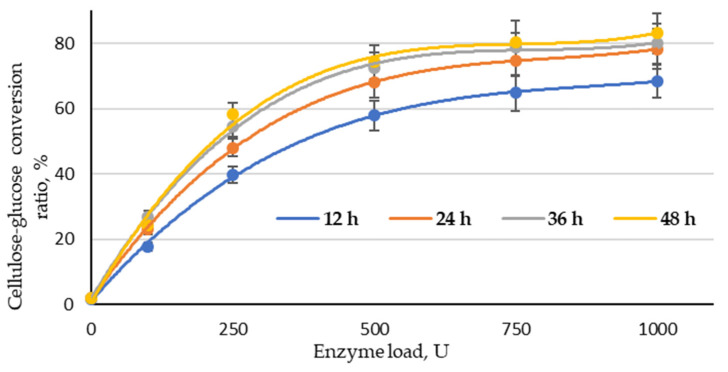
Cellulose–glucose conversion ratio depending on enzyme load and reaction time (LC residue sample treated with an H_3_PO_4_/NaH_2_PO_4_ catalyst mixture at a ratio of 1:1).

**Table 1 molecules-27-07353-t001:** Chemical composition of wood chips used.

Component	Birch Wood Chips	
	Amount	
	% o.d.m.	+/−
Extractives	4.23	0.05
Glucan	35.66	0.92
Arabinan	0.46	0.04
Galactan	1.58	0.43
Xylan	20.68	0.02
Mannan	2.38	0.07
ASL	3.62	0.21
AIL	20.33	0.05
Ash	0.60	0.01
AG	4.11	0.05

o.d.m.—oven-dried mass; ASL—acid-soluble lignin; AIL—acid-insoluble lignin; AG—acetyl groups.

**Table 2 molecules-27-07353-t002:** Effect of H_3_PO_4_/NaH_2_PO_4_ solution on the target products.

Catalyst Ratio	Amount	Birch Wood Chips			
H_3_PO_4_/NaH_2_PO_4_	wt.%	Furfural, % t.p.y.	Furfural, % of o.d.m.	Cellulose, % t.p.y.	Cellulose, % of o.d.m.
0:3	3	39.7 ± 4.5	6.1 ± 0.7	97.1 ± 1.1	35.5 ± 0.4
1:2	3	60.5 ± 3.3	9.3 ± 0.5	98.1 ± 0.6	35.9 ± 0.2
1:1	3	60.5 ± 3.9	9.3 ± 0.6	97.2 ± 1.5	35.7 ± 0.5
2:1	3	62.5 ± 3.3	9.6 ± 0.5	97.9 ± 0.3	35.8 ± 0.1
3:0	3	72.2 ± 2.6	11.1 ± 0.4	83.7 ± 3.3	30.6 ± 1.2

**Table 3 molecules-27-07353-t003:** Dominant carbohydrate contents (g per 15 g of substrate (LC residue sample treated with an H_3_PO_4_/NaH_2_PO_4_ catalyst mixture at a ratio of 1:1)) in enzymatic hydrolysis liquid products depending on enzyme load and hydrolysis time.

	Enzyme Load, U					
	0	100	250	500	750	1000
	Hydrolysis time 12 h					
Cellobiose	0.03 ± 0.00	0.05 ± 0.01	0.10 ± 0.01	0.17 ± 0.05	0.23 ± 0.06	0.27 ± 0.10
Glucose	0.13 ± 0.01	1.35 ± 0.05	3.02 ± 0.09	4.41 ± 0.06	4.94 ± 0.09	5.20 ± 0.12
Xylose	0.23 ± 0.02	0.36 ± 0.03	0.50 ± 0.04	0.60 ± 0.04	0.66 ± 0.07	0.67 ± 0.09
Galactose	0.07 ± 0.01	0.12 ± 0.02	0.16 ± 0.02	0.19 ± 0.02	0.23 ± 0.02	0.21 ± 0.04
Arabinose	0.03 ± 0.00	0.04 ± 0.00	0.05 ± 0.00	0.05 ± 0.00	0.06 ± 0.01	0.07 ± 0.01
	Hydrolysis time 24 h					
Cellobiose	0.03 ± 0.00	0.05 ± 0.00	0.11 ± 0.02	0.20 ± 0.04	0.25 ± 0.04	0.29 ± 0.07
Glucose	0.14 ± 0.02	1.76 ± 0.03	3.65 ± 0.06	5.19 ± 0.12	5.67 ± 0.13	5.95 ± 0.12
Xylose	0.24 ± 0.06	0.37 ± 0.05	0.52 ± 0.04	0.63 ± 0.09	0.67 ± 0.07	0.69 ± 0.05
Galactose	0.08 ± 0.02	0.13 ± 0.01	0.20 ± 0.03	0.23 ± 0.04	0.25 ± 0.06	0.27 ± 0.06
Arabinose	0.02 ± 0.00	0.04 ± 0.00	0.06 ± 0.01	0.08 ± 0.02	0.09 ± 0.02	0.09 ± 0.02
	Hydrolysis time 36 h					
Cellobiose	0.03 ± 0.00	0.05 ± 0.00	0.13 ± 0.04	0.21 ± 0.04	0.28 ± 0.03	0.30 ± 0.05
Glucose	0.15 ± 0.02	2.04 ± 0.12	4.14 ± 0.12	5.51 ± 0.12	5.99 ± 0.14	6.49 ± 0.20
Xylose	0.26 ± 0.03	0.40 ± 0.03	0.56 ± 0.06	0.66 ± 0.06	0.70 ± 0.08	0.67 ± 0.07
Galactose	0.26 ± 0.03	0.12 ± 0.02	0.18 ± 0.02	0.22 ± 0.02	0.21 ± 0.06	0.20 ± 0.03
Arabinose	0.03 ± 0.00	0.05 ± 0.00	0.09 ± 0.03	0.11 ± 0.02	0.12 ± 0.03	0.12 ± 0.02
	Hydrolysis time 48 h					
Cellobiose	0.03 ± 0.00	0.05 ± 0.00	0.14 ± 0.02	0.22 ± 0.06	0.28 ± 0.06	0.31 ± 0.04
Glucose	0.14 ± 0.01	1.85 ± 0.12	4.42 ± 0.12	5.65 ± 0.18	6.12 ± 0.16	6.32 ± 0.18
Xylose	0.25 ± 0.02	0.38 ± 0.06	0.57 ± 0.08	0.66 ± 0.09	0.69 ± 0.07	0.68 ± 0.09
Galactose	0.06 ± 0.00	0.12 ± 0.02	0.22 ± 0.04	0.24 ± 0.04	0.27 ± 0.09	0.23 ± 0.04
Arabinose	0.02 ± 0.00	0.05 ± 0.00	0.09 ± 0.02	0.10 ± 0.02	0.11 ± 0.02	0.10 ± 0.02

## Data Availability

Not applicable.

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
