# Peer review of "Direct Furfural Production from Deciduous Wood Pentosans Using Different Phosphorus-Containing Catalysts in the Context of Biorefining"

_molecules, 2022, doi:10.3390/molecules27217353_

Round 1
Reviewer 1 Report
The manuscript submitted by the authors presented an investigation of furfural production from birch wood chips using autohydrolysis in the presence of phosphoric acid and/or salts. Reasonably high furfural yield was achieved when using phosphoric acid, but not with phosphorous salts. Combination of phosphoric acid and the best salt increased furfural yield compared to using only salt, albeit lower than only using phosphoric acid, with the added benefit of higher cellulose recovery. The resulting cellulose fractions were shown to be likely digestible using a commercial enzyme cocktail, suggesting a potential integration into biorefinery operation.
The overall work is rather structured and standard methods of the field were used. The work can be of interest for the researchers working in the field. However, the manuscript suffered greatly from major flaws in research design which limits its industrial relevance and potential novelty, missing important experiment and technical details, lack of clear presentation of data, as well as extensive typo, style and grammatical error throughout the text. Based on these points a rejection is suggested to the editor. The points are further elaborated below:
Flaw in the research design and lack of novelty
1. The greatest flaw is the use of single condition in the autohydrolysis of lignocellulosic biomass (175oC, 90 min). This limits greatly the potential of generating knowledge and understanding how the variables interact.
2. Additional key flaw related to the previous point which made it worse, especially for the industrial perspective that the work is aiming at, is that the conditions are not realistic for application. Holding such high temperature for 90 min will never be industrially viable in terms of expenditures. Thus, it diminishes any potential yield of furfural and other products gained.
3. The key novelty will have been in the use of the phosphoric salts, however neither reasonably high furfural yield nor elucidation of molecular mechanisms were attained. The aforementioned points 1-2 have also undermined it. Then what remains is the use of phosphorous acid in dilute acid hydrolysis of lignocellulosic biomass and the resulting production of by-products including furfural, which is already well-known in the field.
Missing important experiment and technical details
1. The assay used for cellulase activity is not clear (page 6), the substrate info is missing. The cited method only indicates reducing sugar measurement.
2. Therefore, based on the above, the information on enzyme dosage (page 6-7) is not clear. A dose of e.g. g enzyme/kg dry biomass will have been better to assess digestibility. As it is now, the results are almost meaningless.
3. Missing enzymatic hydrolysis data for other biomass (page 6-7). At least one from autohydrolysis and another using only phosphoric acid will be needed for control. This is imperative to point out the novelty, especially that even though more cellulose is shown to be retained (Table 1) in the 1:1 phosphoric acid : salt treatment, it is important to show if there are improvements in the yield and digestibility to justify the cause.
4. Pressure information is missing in the autohydrolysis process (page 9).
5. There is no address in the purification of the obtained furfural, e.g. how challenging it will be with the other inhibitory by-products (Figure 3).
Lack of clear presentation
1. Overall low quality of figures, see also inconsistent annotations in Figure 4.
2. No indication of replicate in all analyses.
3. No statistical analyses were performed to indicate significance.
4. Missing autohydrolysis legend in Figures 3-5.
5. Missing substrate information in Figure 6 and Table 2.
6. Redundant data in Table 2 and presentation should be in terms of yield.
7. Figure 7 is not really necessary.
Author Response
Dear Reviewer,
Flaw in the research design and lack of novelty:
- The greatest flaw is the use of single condition in the autohydrolysis of lignocellulosic biomass (175ºC, 90 min). This limits greatly the potential of generating knowledge and understanding how the variables interact.
Answer (A): Various process conditions have been studied in a previous publication, cited in the text (Reference No. 10). In that study was concluded that optimal parameters for phosphoric acid catalysed hydrolysis are 175ºC, catalyst amount 3.0 wt.%, 90 min.
- Additional key flaw related to the previous point which made it worse, especially for the industrial perspective that the work is aiming at, is that the conditions are not realistic for application. Holding such high temperature for 90 min will never be industrially viable in terms of expenditures. Thus, it diminishes any potential yield of furfural and other products gained.
A: These conditions are not out of the line of currently utilised furfural production conditions on an industrial scale. Non the less to transfer this approach to an industrial scale, a full economic assessment is vital. It should be stated that this is not in the scope of this publication.
- The key novelty will have been in the use of the phosphoric salts, however neither reasonably high furfural yield nor elucidation of molecular mechanisms were attained. The aforementioned points 1-2 have also undermined it. Then what remains is the use of phosphorous acid in dilute acid hydrolysis of lignocellulosic biomass and the resulting production of by-products including furfural, which is already well-known in the field.
A: Yes, the novelty was to use phosphoric salts in the hydrolysis process as a catalyst. Results indicated that using these salts cannot reach a high conversion ratio of pentosans into furfural but gave an opportunity to preserve cellulose in solid residue. Therefore, we combine phosphoric acid with salt which gives the best results both in furfural outcome and in cellulose amount in solid residue. Such a solution of catalyst has also not been studied before as well we find that this solution of the catalyst allows reduction of the acid amount in the solid residue.
Missing important experiment and technical details:
- The assay used for cellulase activity is not clear (page 6), the substrate info is missing. The cited method only indicates reducing sugar measurement.
A: Measurement of the enzyme activity was performed according to the method of Denault et al. in combination with the method of Dygert et al., cited in the text as references No. 18 and 17.
- Pressure information is missing in the autohydrolysis process (page 9).
A: The pressure was constant (7.9±0.1 bar) in all hydrolysis experiments. This information was added to the text also.
- Therefore, based on the above, the information on enzyme dosage (page 6-7) is not clear. A dose of e.g. g enzyme/kg dry biomass will have been better to assess digestibility. As it is now, the results are almost meaningless.
A: In this case, the dosage of enzyme was U of enzyme/15 g of dry substrate. According to enzyme complex density 1.1 g/mL and determined activity 346 U/mL it can be calculated. As it is well known during the time enzyme activity changes. Therefore, to compare results in future we decided to use such type of expression.
- Missing enzymatic hydrolysis data for other biomass (page 6-7). At least one from autohydrolysis and another using only phosphoric acid will be needed for control. This is imperative to point out the novelty, especially that even though more cellulose is shown to be retained (Table 1) in the 1:1 phosphoric acid: salt treatment, it is important to show if there are improvements in the yield and digestibility to justify the cause.
A: We agree with your suggestion. But this time, to show that it is possible to produce furfural before enzymatic hydrolysis, we presented our preliminary study results of enzymatic hydrolysis. Now, we don't have results where birch woodchips treated with phosphoric acid were used as a substrate. The work will be continued. The enzymatic hydrolysis will be optimised and the results will be more detailed.
- There is no address in the purification of the obtained furfural, e.g. how challenging it will be with the other inhibitory by-products (Figure 3).
A: Not in the scope of this publication.
Lack of clear presentation:
- No indication of replicate in all analyses.
A: All of the analyses were done in triplicates. For each of the obtained results, a standard deviation can be found.
- No statistical analyses were performed to indicate significance.
A: For all of the results standard deviation was calculated.
- Missing autohydrolysis legend in Figures 3-5.
A: The information about the autohydrolysis process data in Figures 3-5 were added to the text.
- Missing substrate information in Figure 6 and Table 2.
A: Added substrate information.
- Redundant data in Table 2 and presentation should be in terms of yield.
A: Authors do not agree with the Reviewer on this comment. This is how we always have shown the results.
- Figure 7 is not really necessary.
A: This Figure (acid hydrolysis reactor system) gives the reader a better understanding and clearer vision of how these hydrolysis experiments were performed. That is why the authors decided to leave Figure 7 in the manuscript.
Reviewer 2 Report
This work is aimed at improving the efficiency of pretreatment of birch chips for the direct production of furfural. For better valorization of birch wood biomass, the authors for the first time used various phosphate salts as a catalyst in order to (1) increase the yield of cellulose and reduce the content of acetyl groups in LC residues in order to mitigate the inhibition of subsequent enzymatic hydrolysis.
The results are of practical interest, but a more detailed discussion of the results obtained is recommended. In particular, there is no comparison of the efficiency of acid and enzymatic hydrolysis with other works. The economic component of the process is also not discussed or compared with other works.
The "Materials and Methods" section should be placed after the "Introduction" section in accordance with the template.
Some comments, including those on grammar, are included in the pdf-file.

Author Response
Dear Reviewer,
All of the errors noted in the pdf have been corrected in the text.
Reviewer 3 Report
The manuscript, entitled “Direct furfural production from deciduous wood pentosans using the different phosphorus-containing catalysts in the context of biorefining” with manuscript ID of Molecules-1877962 for Molecules, reported that the effect of phosphorus-containing salt on furfural production from birch wood chips and cellulose amount in the solid lignocellulosic residue for further processes. Although this work shows some interesting or new insights, there are still many problems to be solved. Please see the detailed comments in the below.(1) The “Keywords” should cover the main content of the article more comprehensively, it is difficult to link the title and the keywords of the manuscript.
(2) Could the authors provide more detailed reasons of why there are no reports on the phosphate salt hydrolysis to produce furfural as well as its effect on enzymatic hydrolysis?
(3) In section “2.4 Hydrolysis – acetyl groups in the LC residue”, why show different results of groups with different amount of catalyst?
(4) It only the effect of catalyst on furfural production have been discussed in whole manuscript? Without temperature and pressure? Why?
(5) It is suggested to add a discussion on the structure-activity relationship between yield and acid amount.
(6) How about the unit of the enzyme load?
(7) The language of the manuscript should be improved.
(8) The conclusion is simple and it should further to analysis and explanation in-depth.
Author Response
Dear Reviewer,
- The “Keywords” should cover the main content of the article more comprehensively, it is difficult to link the title and the keywords of the manuscript.
Answer (A): The Keywords have been changed to: lignocellulose sources, biorefining, furfural, phosphorus-containing catalysts, hydrolysis, and enzymatic hydrolysis
- Could the authors provide more detailed reasons of why there are no reports on the phosphate salt hydrolysis to produce furfural as well as its effect on enzymatic hydrolysis?
A: Probably it is related due to the fact that we use a different approach to the addition of catalysts. The catalyst is evenly sprayed on the surface of biomass before hydrolysis. In conventional hydrolysis, biomass is in a catalyst solution. But in enzymatic hydrolysis phosphate salts provides the required environmental pH and do not inhibit the functioning of the microorganisms.
- In section “2.4 Hydrolysis – acetyl groups in the LC residue”, why show different results of groups with different amount of catalyst?
A: It depends on the degraded biomass amount during the hydrolysis process. With the increase of catalyst amount, more biomass was degraded. So, the ratio of acetyl groups and solid residue increases. Therefore, it looks like the amount of acetyl groups in solid residue increases. But in reality, it is lower. For example, the amount of acetyl groups in the solid residue increases with the increase of phosphoric acid amount in the reaction medium.
- It only the effect of catalyst on furfural production have been discussed in whole manuscript? Without temperature and pressure? Why?
A: Process temperature has been optimized previously and shown in other publication. Cited in the text (Reference No. 10). The system is not closed, therefore the pressure in the hydrolysis reactor depends on the used treatment temperature. In this study treatment temperature was constant (175ºC), therefore pressure in the reactor was constant - 7.9±0.1 bar.
- It is suggested to add a discussion on the structure-activity relationship between yield and acid amount.
A: Not in the scope of this publication.
- How about the unit of the enzyme load?
A: In this case, the dosage of enzyme was U of enzyme/15 g of dry substrate. According to enzyme complex density 1.1 g/mL and determined activity 346 U/mL it can be calculated in g/g. As it is well known during the time enzyme activity changes. Therefore, to compare results in future we decided to use such type of expression.
- The conclusion is simple and it should further to analysis and explanation in-depth.
A: Conclusions have been expanded
Round 2
Reviewer 2 Report
Dear Authors,
not all the recommendations were addressed.
Comparison of the obtained results with other works was not carried out.
The section "Materials and Methods" should be after the section "Introduction" according to the template.
Author Response
Dear Reviewer,
we have added the comparison of some other works with our obtained results.
According to the MDPI Molecules journal template the section "Materials and Methods" have to be after the "Results" section, not after "Introduction".
Reviewer 3 Report
The authors have answered the questions, and this version can be accepted.
Author Response
Dear Reviewer,
thank you for the acceptance of our publication.